# Advances in the Evaluation of Gastrointestinal Absorption Considering the Mucus Layer

**DOI:** 10.3390/pharmaceutics15122714

**Published:** 2023-11-30

**Authors:** Kaori Miyazaki, Akira Sasaki, Hiroshi Mizuuchi

**Affiliations:** DMPK Research Laboratories, Innovative Research Division, Mitsubishi Tanabe Pharma Corporation, 1000 Kamoshida, Aoba-ku, Yokohama 227-0033, Japan; sasaki.akira@md.mt-pharma.co.jp (A.S.); mizuuchi.hiroshi@mp.mt-pharma.co.jp (H.M.)

**Keywords:** mucus layer, mucin, MUC, intestinal absorption, cell membrane permeation

## Abstract

Because of the increasing sophistication of formulation technology and the increasing polymerization of compounds directed toward undruggable drug targets, the influence of the mucus layer on gastrointestinal drug absorption has received renewed attention. Therefore, understanding the complex structure of the mucus layer containing highly glycosylated glycoprotein mucins, lipids bound to the mucins, and water held by glycans interacting with each other is critical. Recent advances in cell culture and engineering techniques have led to the development of evaluation systems that closely mimic the ecological environment and have been applied to the evaluation of gastrointestinal drug absorption while considering the mucus layer. This review provides a better understanding of the mucus layer components and the gastrointestinal tract’s biological defense barrier, selects an assessment system for drug absorption in the mucus layer based on evaluation objectives, and discusses the overview and features of each assessment system.

## 1. Introduction

Oral medication forms are widely used to improve patient quality of life because they are non-invasive and convenient routes of administration. However, oral formulations are absorbed in the gastrointestinal tract and require various gastrointestinal absorption processes, including the movement of the formulation in the gastrointestinal tract, drug elution and dissolution, drug penetration through the intestinal membrane, and transfer into the systemic bloodstream. These processes can alter the pharmacokinetics and efficacy of the drug [1,2,3]. Therefore, an accurate understanding of drug candidate gastrointestinal absorption is critical for efficient drug development.

Because most orally administered drugs have a high biological membrane affinity, the cell membrane permeation of the drug in the intestinal tract occurs by passive diffusion following a drug concentration gradient. Cell membrane permeation follows the pH partitioning hypothesis, particularly for dissociable weak electrolyte compounds. Therefore, a drug’s lipid solubility and the ratio of its presence at the absorption site influence drug absorption [4]. Drug absorption is also influenced by gastrointestinal tract physiological characteristics, as evidenced by the efflux transporter P-glycoprotein, which acts as an absorption barrier by transporting the drug from the intestinal cells to the lumen [5].

In addition to drug transporters, the gastrointestinal mucus layer adjacent to the intestinal mucosa has been identified as a physiological factor influencing drug absorption in the gastrointestinal tract. The mucus layer is a highly viscous liquid phase that covers the gastrointestinal epithelium and acts as a physical barrier to pathogens and commensal bacteria entering the mucosa [6,7,8]. Because drugs must pass through the mucus layer to reach the epithelial cell membrane surface, it may also be an important regulator of drug absorption in the gastrointestinal tract. The mucus layer is composed of proteins, lipids, sugar chains, and water, and is thought to act as a barrier to drug absorption through the following mechanisms: (1) drug–mucus component interactions (drug physical properties, such as hydrophobicity and charge), and (2) molecular size selectivity (due to the network structure of the mucus components) [9,10,11] (Figure 1). Therefore, the influence of the mucus layer on drug membrane permeation is becoming increasingly important. However, unlike drug transporters [12,13,14] with a systemic evaluation established in the preclinical phase, a systematic evaluation system for the mucus layer has not been established. This is presumably because, unlike transporters comprising a single protein, the mucus layer is composed of complex components, and the resulting complex chemical and physical properties render constructing an evaluation system challenging. Therefore, drug development evaluation systems focusing on the mucus layer are still in the early stages.

Recently, to address undruggable targets, the pharmaceutical industry has been actively developing oral mid-molecular-weight drugs, such as cyclic peptides and targeted protein degraders, from existing simple small molecules [15,16,17,18]. Because of properties such as high lipophilicity, large molecular weight, low solubility, and high protein binding, there is concern that mid-molecular-weight drugs may interact with the mucus layer. When added to extracted mucus components, cyclosporin A causes aggregation [19]. In addition, mid-molecular-weight formulations are often specially formulated to overcome low membrane permeability and solubility (e.g., self-micro-emulsifying drug delivery systems and nanoparticle formulations) [20,21,22]. Maria et al. found a 16-fold increase in mucus adhesion compared to regularly modified nanoparticles and a 7.2-fold increase in AUC for oral administration compared to free octreotide solution, using modified nanoparticles that promote mucus adhesion [23]. When assessing drug absorption using mid-molecular-weight formulations and associated special formulation techniques, the complex and varied interactions of the mucus layer become increasingly important. Consequently, this review provides an overview of the various drug interactions with mucus layer constituents and examines the influence of the mucus layer on drug absorption. Furthermore, advances in cultured cell technology and engineering techniques have led to the development of systems that closely mimic the ecological environment and have been used to assess gastrointestinal absorption. Novel assessment techniques focusing on the mucus layer and the advantages and disadvantages of the various models available are discussed to help select appropriate models for assessing the influence of the mucus layer on gastrointestinal drug absorption.

## 2. Gastrointestinal Mucus

Gastrointestinal mucus is secreted at 10 L per day and contains mucin (5%), lipids (37%), proteins (39%), DNA (6%), and other components by dry weight (%, *w*/*w*) [24] (Figure 1). For absorption in the gastrointestinal tract, the drugs must pass through the mucus layer, permeate the gastrointestinal membrane, and enter the systemic circulation. Therefore, the first barrier to gastrointestinal membrane permeation is the mucus layer, consisting of two layers: an inner, tightly adherent layer and an outer, loosely adherent layer that undergoes repeated secretion and turnover [25]. The two-layer structure maintains the homeostasis of the intestinal flora and protects the intestinal epithelium from intestinal pathogens, as the inner layer is mostly sterile, while the outer layer serves as a habitat for bacteria [26]. When fluorescent beads of 1 µm diameter are added to the structure, the beads penetrate the outer layer, but do not penetrate the inner layer [27]. Mucin molecules play an important role in the maintenance function and as components of this structure [26,28]. Mucus layer thickness in rats, measured using a micropipette before and after aspiration removal, was the thickest in the ileum and thinnest in the jejunum, ranging from 100 to 800 µm. In contrast, the thickness of the human gastrointestinal mucus layer ranges from 10 to 750 µm [29]. The exact thickness of the mucus layer in humans remains controversial because human tissues are generally derived from patients whose mucosal epithelium is in poor health or whose mucus secretion is abnormal. Although replicating the complex properties of the mucus layer, such as its multiple components, bilayer structure, intestinal flora, and inconsistent thickness, is challenging, an appropriate evaluation system that matches the desired results and how to incorporate them must be selected.

## 3. Components of the Mucus Layer

### 3.1. Mucin

Mucin, the molecular entity of the mucus layer, is a highly glycosylated glycoprotein. The mucus layer is a charged hydrophilic layer due to the mucin sugar chain’s negative charge and water-holding capacity [29,30,31]. Mucins are classified into membrane-bound (MUC1, MUC3A, MUC3B, MUC4, MUC12, MUC13, MUC14, MUC15, MUC16, MUC17, MUC20, MUC21, and MUC2) and secretory (MUC2, MUC5AC, MUC5B, MUC6, and MUC19), based on their functional characteristics and localization [7,32,33]. Transmembrane mucins are plasma membrane-anchored membrane proteins with a single transmembrane domain, an amino-terminal extracellular region densely decorated with sugar chains, and a carboxy-terminal intracytoplasmic tail. Because of the highly glycosylated core protein, membrane-bound mucins have a test-tube brush, and MUC1, MUC3A, MUC3B, MUC4, MUC12, MUC13, MUC15, MUC17, MUC20, and MUC21 are expressed in the gastrointestinal tract [31,34]. Compared with normal MCF7 cells, MCF7 cells stably expressing MUC1 with the parts involved in signaling removed showed a 150-fold increase in resistance to the fat-soluble drug paclitaxel and a 2.7-fold increase in cell adhesion water content [35] (Figure 2), suggesting that MUC1 acts as a hydrophilic barrier against anticancer drugs and contributes to chemotherapy resistance by limiting lipophilic drug membrane permeation. Secretory mucins have a large glycosylated PTS domain, a light glycosylated carboxy- and amino-terminal region rich in cysteine, and form a gel [31,36,37]. Because of this geometry, secretory mucins are retained as a mesh-like barrier surrounding the plasma membrane. MUC2, MUC5AC, and MUC6 are expressed in the gastrointestinal tract. Particle mobility measurements in sputum mucus from patients with cystic fibrosis (CF) decreased with increasing particle size [38]. The high content of MUC5AC in sputum from patients with CF supports the idea of size filtering by secreted mucins [39]. Because mucins have many subtypes that differ in properties and molecular size, it is important to focus on mucin molecular species differences when studying the microscopic environment of the mucus layer.

### 3.2. Glycans

Many proteins exist as glycoproteins, which are made up of glycans bound covalently to amino acids via glycosylation; these glycans play important roles in biological tissue formation and defense [40]. Typical forms of glycosylation found on secreted or membrane-bound proteins are *N*-linked (to asparagine) and mucin-type *O*-linked (to serine or threonine) glycosylation [41]. Mucins are glycoproteins with a unique structure in which a series of *O*-linked glycans derived from N-acetylgalactosamine are attached to serine and threonine residues in tandem repeat domains [9]. Glycans contribute to water and charge retention in the mucus layer [6,29,30,31]. Mucin glycosylation and expression distribution differ by subtype. In mice, MUC5AC is expressed in the upper gastrointestinal tract, primarily in the stomach, and approximately half of its glycans are neutral, with many monosulfated glycans but few fucosylated or sialylated glycans. In contrast, MUC2 is predominantly expressed in the lower gastrointestinal tract, primarily in the large intestine, dominated by fucosylated glycans and negatively charged sialylated and sulfated glycans [42]. Rats showed similar gastrointestinal site-specific glycosylation [43]. Structural characterization of oligosaccharides released from purified human mucin by gastrointestinal sites revealed that fucosylated glycans were mainly detected in the small intestine, whereas sulfated glycans were mainly detected in the distal colon [44]. This suggests that there are species differences in the glycosylation of mucins between rodents and humans. In addition to the mucosal layer thickness and mucin subtype, glycan characteristics depend on the gastrointestinal tract site.

In endogenous MUC1-high expressing Capan-1 cells, inhibiting enzyme O-glycosylation with glycan synthesis inhibitors leads to a decrease in MUC1 glycans and an increase in 5-fluorouracil incorporation into genomic DNA [45,46] (Figure 3). This suggests a link between MUC1 glycan levels and the 5-fluorouracil cellular uptake’s therapeutic effect. Furthermore, the thickness of the cell surface water layer differs between cells transgenic for MUC1 and MUC13, which have different lengths of glycosylated extracellular domains [35]. These findings suggest that the electronegativity of the mucin glycan moiety or the water layer thickness retained by the glycans may influence drug permeation through the plasma membrane. The mucin glycosylation region is extensive, and glycans are important in drug cell membrane permeation because they influence water retention and charge-bearing.

### 3.3. Lipids

Lipids are one of the most important nutrients; during absorption in the small intestine, they assist in the absorption of fat-soluble vitamins [47]. In living organisms, lipids are homeostatic because they are essential components of cell membranes, where cell signaling and membrane proteins are prepared [48]. Endogenous lipids in mucus include cholesterol, ceramides, palmitic acid, stearic acid, oleic acid, linoleic acid, other free fatty acids, and polar lipids [24]. Sputum analysis from patients with CF confirmed that the lipids in the purified mucin fraction were complexed with glycoproteins [49]. Lipid extraction from canine gastric mucus glycoproteins reduced viscosity by 80–85%, and removing covalently bound fatty acids further reduced the viscosity of defatted glycoproteins by 39% [50]. Fluorescence probing of the hydrophobicity of gastric mucus glycoproteins revealed that some fatty acids were covalently bound to mucins in the protein domain, whereas most were adsorbed by hydrophobic bonds [51]. These findings demonstrate that lipids form hydrophobic and covalent bonds with mucins in the mucus layer and regulate mucus viscoelasticity. Yildiz et al. showed that when microspheres were injected into the duodenum of rats 1 h after the oral administration of soybean oil, the transport rate of the microspheres was reduced by a factor of 10 compared with that in the control group [52]. Furthermore, the addition of lipids associated with food and drug delivery systems increased the elasticity of the mimetic mucus. These results suggested that the amount of dietary lipids in the gut and those used in drug delivery systems must be considered when transporting drug carriers through the mucus layer. However, excessive lipid intake is known to affect the function of the intestinal barrier and mucus layer. Gulhane et al. showed that a prolonged high-fat diet induces a decrease in mucosal barrier function due to goblet cell differentiation, a decrease in Muc2, a loss of tight junction proteins, and an increase in serum endotoxin levels [53]. Compared to the physiological environment, assessments using cultured cells or commercially available purified mucins may result in fewer lipids or their removal during the production process, which may not reflect the in vivo environment. Therefore, the viscosity of the mucus layer and its interaction with lipid components should be considered.

## 4. In Vivo Experimental Methods

### 4.1. Mucolytic Agent

Mucolytic agents such as dithiothreitol (DTT) and N-acetylcysteine (NAC) are used in experimental animals to assess the effect of the mucus layer on gastrointestinal drug absorption. Mucolytic agents work by cleaving disulfide bond cross-links in secretory mucin and disrupting the intermolecular network structure [31,54]. The intranasal administration of fluorescein isothiocyanate (FITC)–dextran with NAC in rats resulted in a 4-fold increase in AUC compared to normal FITC–dextran, confirming that mucus layer removal by NAC improves drug mucosal permeability [55]. Using an in situ single-pass perfusion technique, Masaoka et al. demonstrated that the intestinal membrane permeability of griseofulvin was significantly increased in jejunum pre-treated with DTT for 30 min [56]. Similarly, when DTT was pre-activated in vitro for 10 min, the intestinal membrane permeability of griseofulvin and antipyrine was increased in the proximal part of the small intestine of the duodenum and jejunum, but not in the distal part of the small intestine [57] (Figure 4). Mucus layer removal increased the drug’s mucosal permeability, indicating that the effect varies depending on the compound administered and the gastrointestinal tract site. Although in situ single-pass perfusion and in vitro sac techniques are useful for assessing the site-specific absorption of the mucus layer in the gastrointestinal tract, the concentration of local mucolytic agents can be higher than with conventional administration. If site-specific absorption is assessed as an effect of the mucus layer throughout the gastrointestinal tract, oral administration to animals is preferable; however, it must be ensured that the mucus layer remover reaches the target gastrointestinal absorption site and does not affect compound solubility or stability.

### 4.2. Mucin Knockout/Mucin-Deficient Mice

Mucin knockout (KO) mice are frequently used to study the function and characteristics of mucin molecules in vivo. In terms of membrane permeation, Muc1−/− mice have a decreased uptake and absorption of cholesterol in the gastrointestinal tract compared to normal mice [58]. FITC–dextran was detected in the plasma of Muc2−/− mice but not in normal mice after oral administration [59]. These findings suggest that MUC1 and MUC2 play important roles in compound uptake and absorption in the gastrointestinal tract. The effect of MUC1 on cholesterol uptake in the gastrointestinal tract is particularly intriguing, as the mucin molecule facilitates rather than hinders the compound’s gastrointestinal absorption. Mucus layer thickness in the stomach of Muc1−/− mice was reduced compared to wild-type mice, and MUC5AC was the major component of the mucus layer [60]. Thus, the formation of a macromolecular complex of membrane-bound and secreted mucin anchored on the intestinal epithelial cell membrane may help to stabilize secreted mucin on the intestinal surface, and mucins may interact between subtypes. The mucin KO mice allow for the assessment of the effect of mucin molecules on the ecological environment; however, some mucins also affect inflammatory mechanisms and tumor resistance [61,62,63,64,65]. For example, the microscopy of the colon of Muc2−/− mice revealed mucosal thickening, increased proliferation, and surface erosion, indicating damage to areas other than the mucus [65]. Therefore, KO mice may have altered intestinal histology and environment, leading to different pharmacokinetic data than normal mice.

## 5. In Vitro Experimental Methods

### 5.1. Mucus-Secreting Cells

Caco-2 cells do not have a mucus volume corresponding to the human intestinal tract due to the lack of mucus-secreting cells and low mucus-producing capacity, and the amount of mucus in model cells and in the in vivo environment differs in the literature [66,67,68]. To compensate for this drawback, a co-culture system of Caco-2 and HT-29 cells derived from mucus-secreting human colon cancer has been used to simultaneously assess mucus layer permeabilization through the gastrointestinal membrane [69,70]. When Caco-2 intestinal epithelial cells were co-cultured with HT29-MTX cells, mucus formation was observed, and a 40–80 µm thick mucus layer formed in the transwells, depending on the amount of HT29-MTX cells [71]. HT-29 cells treated with fluorouracil and methotrexate can generate HT29–5-fluorouracil cells expressing high levels of MUC2 and MUC4 or HT29–MTX cells expressing high levels of MUC3 and MUC5AC [72]. Although this does not allow for a complete subtype selection, it does allow for some examination of mucin subtypes without genetic recombination manipulation. HT29-MTX cells were used not only as a simple active pharmaceutical ingredient and to evaluate the effect of nanoparticle formulations on the mucus layer [73]. In cells with incomplete mucus formation, there was no significant difference in cellular uptake between the nanoparticle formulations; however, in cells with confirmed mucus formation, nanoparticle formulations with significantly higher cellular uptake were found. The use of HT29-MTX cells made it possible to select a nanoparticle formulation that considered the effect of the mucus layer. However, because the culture system contains two types of cells, the culture conditions must be controlled, and the P-glycoprotein substrate has a different membrane permeability from that of Caco-2 cells because HT29 cells do not express P-glycoprotein [68].

In recent years, cell culture systems have advanced remarkably, particularly in terms of induced pluripotent stem (iPS) cells, which differentiate into functional intestinal cell-like cells and are used in pharmacokinetic studies [74,75,76]. MUC2 mRNA expression levels in iPS cells are equivalent to those in human intestinal cells and can be used for membrane permeabilization studies [77]. Adding indomethacin, a non-steroidal anti-inflammatory drug, to iPS cells decreases, whereas adding the mucoprotective agent rebamipide increases MUC2 expression levels [78]. Furthermore, indomethacin causes mucosal damage and increases the membrane permeability of FITC–dextran, whereas rebamipide restores membrane permeability. These findings indicate that the barrier function of the mucus layer can also be assessed in iPS cells and that iPS cells are steadily evolving into cells capable of the pharmacokinetic evaluation of the mucus layer. The presence of goblet cells, MUC2 production, and mucin granules in iPS cell-derived intestinal organoids has also been confirmed [78,79,80]. Various human iPS cell-derived intestinal organoids with colon-like and small intestine-like properties have also been generated, and could provide an evaluation system closely reflecting in vivo location characteristics [79]. While iPS cell-derived intestinal organoids are used for cellular drug uptake studies, the use of intestinal organoids as a membrane permeability test is difficult due to their shape [80].

### 5.2. Extracted and Mimicked Mucus

Mucus interaction and mucus layer permeability have been studied using mucus extracted from patients with CF, cultured cells, and animals [19,24,81,82,83]. Studies using commercially available animal-derived mucins (porcine and bovine) have become the mainstream due to variations in extraction methods, inter-individual differences between animals, the pathological conditions of patients, and easy availability [84]. As mentioned above, mucus has a complex structure and contains various components, rendering mucus extraction without damage and its reproduction difficult. The biosimilar mucilage had a viscosity 100 times lower than the elastic properties of porcine intestinal mucilage [85], presumably because commercial mucins are fragmented during purification, and contaminating proteins can alter their gel-forming capacity, as it is difficult to isolate and purify mucins while preserving their natural structure and avoiding contaminants [86,87,88]. However, specific criteria, such as viscosity, can be set for mimicking. Biosimilar mucus, that can be used with cultured cells by adding polymeric thickeners and adjusting the lipid content, has been developed using information from comprehensive component analyses of porcine gut mucus [75,89] (Figure 5). When the permeability of various model compounds in Caco-2 cells with and without this mucus was compared, the permeability was reduced by 1.2- to 6.8-fold in biosimilar mucus-added cells, with the effect being most pronounced for the lipophilic drug testosterone [85]. These results suggest that biosimilar mucus could be used as a barrier function assessment system for cell membrane permeability.

Biosimilar mucus is a high-throughput assay that can be measured without biological materials because commercial products eliminate the need for mucus extraction. In particular, the parallel artificial membrane permeability assay allows many compounds and formulations to be studied simultaneously, and the membranes used do not require the use of biological components [90,91]. The transwell mucus diffusion model with mucus added to the insert is a high-throughput evaluation system that allows many compounds and formulations to be studied simultaneously. Friedl et al. evaluated the effect of size and chemical composition of self-emulsifying drug delivery systems on mucus layer permeability through a transwell model incorporating intestinal mucus [92]. The mucus permeability of fluorescein diacetate incorporating Cremophor RH40 increased in a concentration-dependent manner, indicating Cremophor RH40 as a promising excipient. Extracted and mimetic mucus evaluation systems can also help select excipients by assigning a rank order based on mucus influence.

### 5.3. Microphysiological Systems (MPS)

MPS, such as organ-on-a-chip based on microfluidic devices, are attracting attention as a new cell culture platform. Culture environments that are difficult to replicate with conventional cell culture systems, such as mechanical stimulation, 3D extracellular matrix environments, and cell–cell interactions, can be constructed using these microfluidic devices [93,94,95,96]. Caco-2 cells have been cultured on microfluidic devices and used to assess drug cell membrane permeability under conditions that mimic the intestinal environment [97,98,99]. Although MUC2 was not expressed in Caco-2 cells cultured in normal transwell inserts, it was expressed in Caco-2 cells cultured on microfluidic devices [100,101].

Techniques for cultivating intestinal cell organoids-on-chips have also been developed. After dissociation, Sontheimer-Phelps et al. cultured primary human colon epithelial cells as organoids-on-a-chip [102]. Organoid cultures on transwell inserts and Matrigel demonstrated little goblet cell differentiation, whereas organoid cultures on the Colon Chip revealed that approximately 15% of epithelial cells differentiated into goblet cells. This matched the percentage of goblet cells found in human colon samples. Furthermore, when colon epithelial cells were cultured in the Colon Chip underflow, the mucus layer accumulated to approximately 570 µm and was a bilayer structure with an inner layer impermeable to fluorescent beads and an outer layer permeable to fluorescent beads (Figure 6). This result reflects in vivo mucus layer characteristics that could not be replicated in normal cultured cells. In addition, the transparency of the Colon Chip method allows the secretion and accumulation of the mucus layer to be monitored over time under microscopic imaging, allowing the thickness to be observed without the cumbersome and mucus-damaging risk of using an alcohol-based fixing solution.

Similarly, duodenum-derived organoids-on-a-chip were generated, and genome-wide transcriptional profiling between human duodenal tissue, duodenal-on-a-chip, and Caco2 cell-on-chips showed that duodenal-on-a-chip closely resembled duodenal tissue [103,104] (Figure 7). Furthermore, the duodenal-on-a-chip had a 10-fold higher MUC2 expression than the Caco2 cell-on-a-chip, suggesting that combining microfluidic devices and organoids can effectively mimic the in vivo environment [104] (Figure 7). In addition, the effluent from the channels can be collected, allowing the sampling of secreted substances. Microchannels lined with endothelial cells, which are not present in normal organoid cultures, may allow the assessment of the mucus layer and cell membrane permeation of drugs.

In addition, mucus-on-a-chip has been developed to evaluate the interaction between nanoparticles and the mucus layer without the use of cells or animal samples [105,106]. Wright et al. developed a device that clearly visualizes the mucosal adhesion behavior of chitosan nanoparticles, a phenomenon that is difficult to observe with standard permeation models. In contrast, mesoporous silica and poly (lactic-co-glycolic) acid showed mucosal permeation, demonstrating that specific mucus-nanoparticle binding makes a significant difference in the permeation pattern [106]. This device is useful because it eliminates cytotoxic effects, which are a concern in nanoparticle evaluation, and allows observation of mucus adhesion behavior.

However, the on-chip approach of assessing the mucus layer has only been used in a few cases, and a detailed characterization of the secretory mucus layer remains lacking, warranting further research in the future.

## 6. Conclusions and Future Perspectives

As formulations become more complex and drug candidates have higher molecular weights, research into the barrier function of the mucus layer is becoming increasingly important, as the mucus layer is more susceptible to intestinal drug absorption. The advantage of in vivo evaluation is that the mucus layer’s contribution can be assessed in a biological environment. As previously stated, the effects of the altered intestinal environment in mucin KO mice must be considered, as the KO impairs barrier function and alters immune and signaling functions. Furthermore, for mucolytic agents, the effects of pH and concentration in the gastrointestinal tract must be considered to design dosages that do not impair efficacy and to set concentrations that account for gastrointestinal tract toxicity. However, it is not suitable for analyzing the contribution of each component and must be avoided from an animal welfare point of view as the studies are conducted using laboratory animals.

In vitro evaluation systems consider animal welfare perspectives and provide a high degree of design freedom. It is possible to calculate a drug’s mucus layer permeability alone using extracted or mimetic mucus, and by combining it with cell membrane permeation studies, gastrointestinal permeation can also be evaluated considering both the mucus layer and cell membrane permeation processes. The addition of mucus to Caco-2 cells and PAMPA, which are commonly used in conventional drug discovery research, could allow the use of existing platforms and enable high-throughput testing. Furthermore, in vitro evaluation of the mucus layer has advanced, and the two-layer structure of the mucus, which could previously only be reproduced in vivo, can now be reproduced using MPS, which may further improve in vivo reproducibility in vitro.

The development of evaluation techniques has assisted in assessing the influence of the mucus layer on gastrointestinal drug absorption from various perspectives, depending on the evaluation system used. To assess the mucus layer having complex components, the strengths and weaknesses of the assessment model must be considered to select an appropriate method. This review provides an important understanding of newly developed mucus layer assessment techniques, which will facilitate the analysis and approval of novel drugs. Furthermore, many examples of the mucus layer’s influence have been investigated in various in vivo and in vitro assessment systems; however, the only use of the mucus layer for extrapolation to humans and the prediction of human pharmacokinetic data has been as a non-stirred water layer. We believe that these findings will be used in the future to improve data prediction accuracy by incorporating the mucus layer into prediction models.

## Figures and Tables

**Figure 1 pharmaceutics-15-02714-f001:**
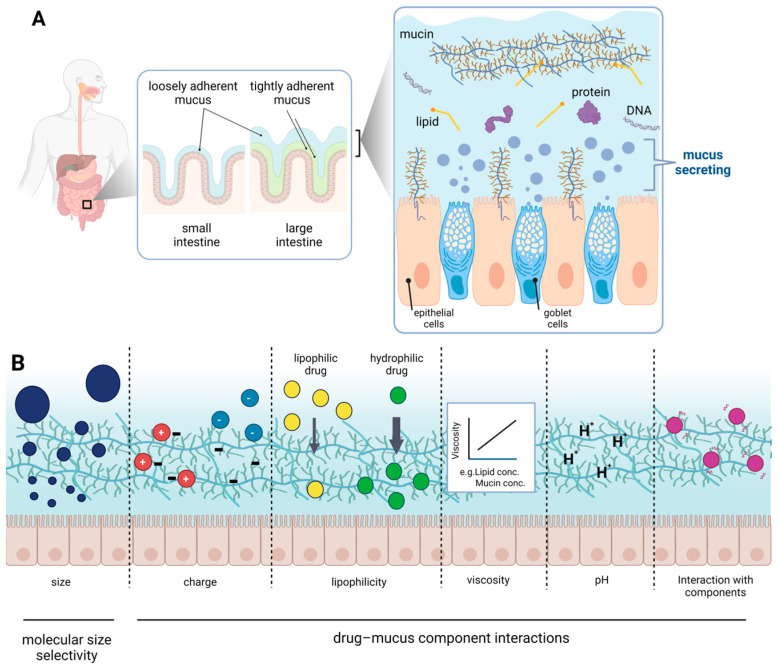
Illustration of gastrointestinal mucus (**A**) and its effect on drug diffusion (**B**). Created with BioRender.

**Figure 2 pharmaceutics-15-02714-f002:**
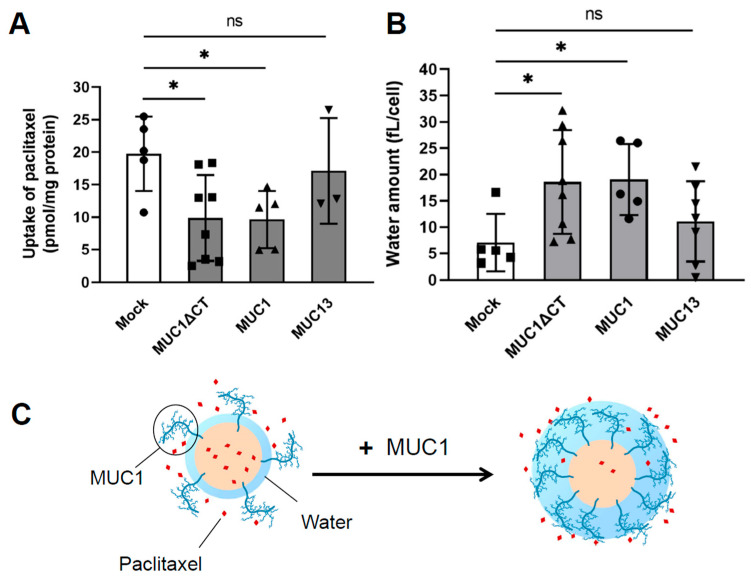
Inhibitory effect of MUC1 cell-associated water on cell accumulation of paclitaxel and its schematic representation. (**A**) The uptake experiment was performed with paclitaxel (10 µM, 60 min). (**B**) Cell-adhered water volume. (**C**) Schematic representation of (**A**,**B**). All symbols represent independent experiments. ns: not significant, * *p* < 0.05 compared with the control (mock cells). Adapted with permission from Ref. [35]. Copyright 2022 American Society for Pharmacology and Experimental Therapeutics.

**Figure 3 pharmaceutics-15-02714-f003:**
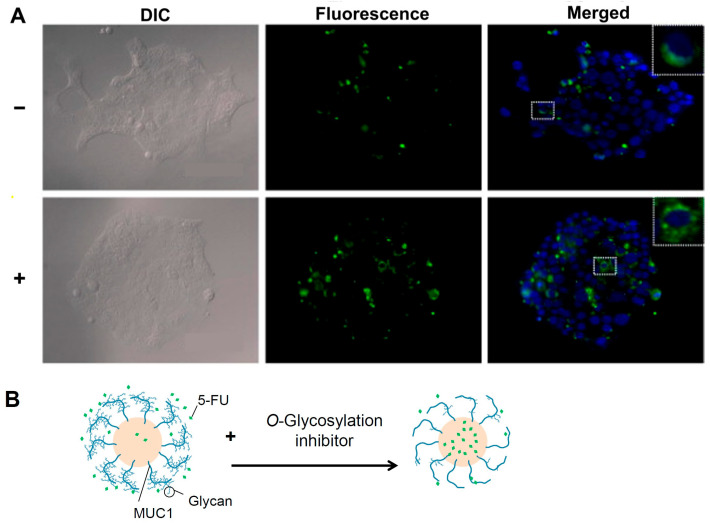
Effects of inhibition of O-glycosylation on 5-FU uptake by Capan-1 cells and its schematic representation. (**A**) 5-FU staining following 1 h of 5-FU exposure in the presence (+) or absence (−) of benzyl-α-GalNAc. The 5-FU was stained with 5-FU antibody (green). The cell nucleus (blue) and 5-FU-antibody (green) were observed using fluorescence microscopy. Scale bars = 100 μm. (**B**) Schematic representation of (**A**). Adapted with permission from Ref. [45]. Copyright 2022 Elsevier.

**Figure 4 pharmaceutics-15-02714-f004:**
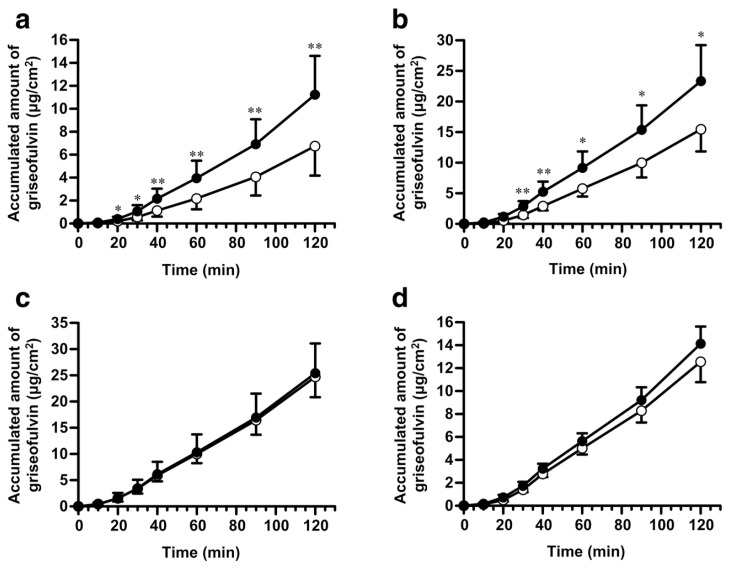
Intestinal permeation of griseofulvin after DTT at 10 mM for 10 min (filled circles) and untreated (open circles) in rat intestinal sacs prepared from (**a**) duodenum, (**b**) jejunum, (**c**) ileum, and (**d**) colon. * *p* < 0.05, ** *p* < 0.01 compared with the control condition. Reproduced with permission from Ref. [57]. Copyright 2019 Springer Nature.

**Figure 5 pharmaceutics-15-02714-f005:**
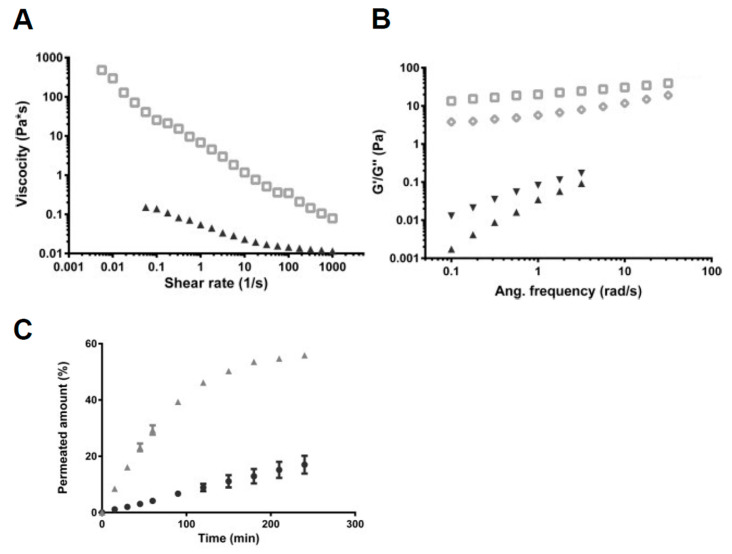
Comparison of native porcine intestinal mucus (PIM) and mucus mixture. (**A**) Flow curves of the PIM (open squares) compared with mucus mixture (triangles). (**B**) Elastic (G′) and viscous (G″) modulus of extracted mucus; G′ (open squares) and G″ (open diamonds) of PIM and G′ (triangles) and G″ (inverted triangles) of the mucus mixture. (**C**) Cumulative amount of testosterone resulting from its penetration through the Caco-2 cell monolayer in the absence (triangles) or presence (circles) of biosimilar mucus. Reproduced with permission from ref [85]. Copyright 2014 Elsevier.

**Figure 6 pharmaceutics-15-02714-f006:**
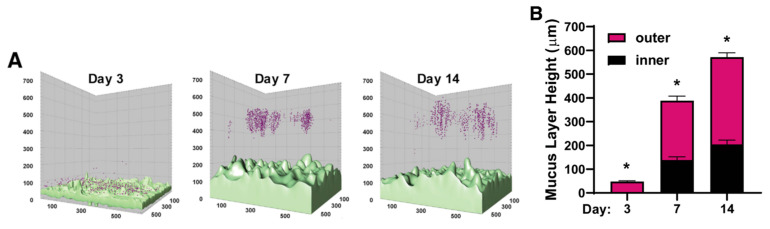
Mucus bilayer of human Colon Chip. (**A**) A pseudocolor 3-dimensional reconstruction of confocal images of the Colon Chip perfused with fluorescent beads (magenta). Cells were stained with calcein AM (light green). (**B**) Quantification of (**A**). * *p* < 0.05 for the inner and outer layers. Reproduced from ref [102].

**Figure 7 pharmaceutics-15-02714-f007:**
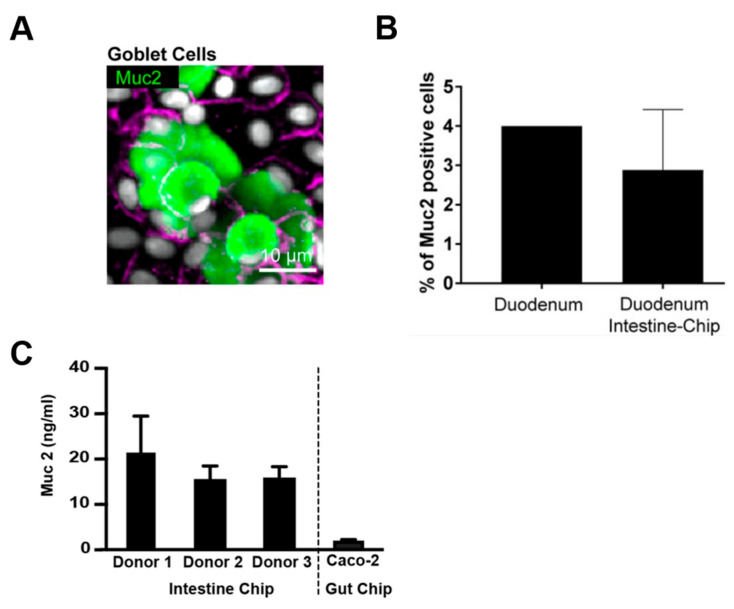
Comparison of Muc2 expression levels in a duodenal intestinal chip, human duodenum, and Caco-2 cell. (**A**) Confocal fluorescence micrographs showing the presence of goblet cells stained with anti-mucin-2 (green). Scale bars = 10 μm. (**B**) Quantification of (**A**). Reproduced from ref. [103]. (**C**) Apical secretions were collected in the effluent of the epithelial channel of the intestine chip generated from organoids and a human Caco-2 cell line-based gut chip. Reproduced from ref [104].

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
