# Peer review of "Advances in the Evaluation of Gastrointestinal Absorption Considering the Mucus Layer"

_pharmaceutics, 2023, doi:10.3390/pharmaceutics15122714_

Round 1

Reviewer 1 Report

Comments and Suggestions for Authors

The theme of this review is novel and interesting, and it is also one of the current research hotspots. My suggestions are as follows.

1.  Approximately 50 references for this review were published 10 years ago. In order to better present the current research status, it is recommended that the authors replace the old literature with updated literature.

2. In my opinion, the in vitro evaluation of nanoformulations is one of the current research focuses in this field, and it is also the most important and attractive part of this review. The authors have listed some examples of in vitro experiments of nanoformulations in the review, but it still appears insufficient. It is recommended that the authors add more the latest in vitro evaluation studies of nanoformulations or provide a separate section to describe them.

Author Response

We thank the reviewer for these helpful comments. 
Here is a point-by-point response to the reviewers’ comments and concerns.

Comment 1:  Approximately 50 references for this review were published 10 years ago. In order to better present the current research status, it is recommended that the authors replace the old literature with updated literature.

Response: Unfortunately, much of the literature on the composition of the mucus layer has not progressed very far, but we have cited as much as possible the newer literature (also related to comment 2) and removed the older literature from the citation.

Comment 2: In my opinion, the in vitro evaluation of nanoformulations is one of the current research focuses in this field, and it is also the most important and attractive part of this review. The authors have listed some examples of in vitro experiments of nanoformulations in the review, but it still appears insufficient. It is recommended that the authors add more the latest in vitro evaluation studies of nanoformulations or provide a separate section to describe them.

Response: We agree with the reviewer's comments. We have added a note on the latest in vitro evaluation studies of nanoformulations considering the mucus layer, with a particular focus on evaluation systems where nanoformulations can be assigned a rank order (section5.1., line 267-273 and section5.3., Paragraph 4)

Reviewer 2 Report

Comments and Suggestions for Authors

Present work makes a review on the components of the gastrointestinal mucus layer and the in vitro and in vivo current methods to evaluate the drug adsorption. This review focuses on the importance of exhaustive studies of drug adsorption for good drug design, which consequently leads to greater efficacy in treatments. It compiles useful information for the scientific community, but could improve several aspects:

- In vitro and in vivo always in italics

- More figures must be added to the main text. It is too dense and schemes will help readers to follow the argumentation. Maybe figure 1 must be completed with some mechanisms of generation or interaction of the mucus with the drugs.

- Best descriptions in the components sections are required. Authors only define nature of mucin. They must do the same with other components.

- I know that it is difficult, but authors must find some synonims for gastrointestinal layer, or even "layer" word, to avoid repetitions in the same sentence.

After these modifications, paper can be accepted in Pharmaceutics

Author Response

We thank the reviewer for the careful review. Here is a point-by-point response to the reviewers’ comments and concerns.

Comment 1: In vitro and in vivo always in italics.

Response: We agree with this and have incorporated your suggestion throughout the manuscript. The words "in vitro" and "in vivo" have been changed to italics.

Comment 2: More figures must be added to the main text. It is too dense and schemes will help readers to follow the argumentation. Maybe figure 1 must be completed with some mechanisms of generation or interaction of the mucus with the drugs.

Response: We agree. As requested, a diagram of the mechanism of mucus and drug production or interaction has been added to Figure 1.

Comment 3: Best descriptions in the components sections are required. Authors only define nature of mucin. They must do the same with other components.

Response: We agree. As you pointed out, the content was biased towards "3.1. Mucin", so we fleshed out the content of Sections "3.2. Glycans (line 137-143, 150-154)" and "3.3. Lipids (line 175-178, 188-198)".

Comment 4: I know that it is difficult, but authors must find some synonims for gastrointestinal layer, or even "layer" word, to avoid repetitions in the same sentence.

Response: While we agree with the importance of avoiding repetition of synonyms, the words "layer" and "gastrointestinal layer, " were not changed due to the lack of appropriate synonyms that are commonly used.

Reviewer 3 Report

Comments and Suggestions for Authors

pharmaceutics-2710629

Advances in evaluation of gastrointestinal absorption containing the mucus layer

The manuscript by Miyazaki et al. provides a comprehensive understanding of newly developed mucus layer assessment techniques and how they facilitate the analysis and approval of novel drugs. The authors have well structured the manuscript and presented sufficient information. I suggest the authors kindly consider some minor points below to improve this manuscript prior to publication.

1. Paragraph 1/ Introduction needs citations to support.

2. Please avoid citing too many references simultaneously, such as [12-18] (line 58).

3. “MUC1–/– mice” and “MUC2–/– mice”: Please kindly check and revise the terms appropriately if necessary. Generally, MUC1 refers to a protein, and Muc1 refers to a gene.

4. Abbreviations are required when they are reused. However, some abbreviations, such as NSAID, were introduced but not used elsewhere.

Author Response

We thank the reviewer for the careful review and the positive comment. Here is a point-by-point response to the reviewers’ comments and concerns.

Comment 1: Paragraph 1/ Introduction needs citations to support.

Response: Agree. As suggested, we have added References to Paragraph 1/Introduction, such as [1-3] (line 28).

Comment 2: Please avoid citing too many references simultaneously, such as [12-18] (line 58).

Response: We agree. We have, accordingly, minimized the number of documents cited, such as [12-18] to [15-18] (line 58), [54-59] to [61-65] (line 248), [19, 24, 74–78] to [19, 24, 81–83] (line 296).

Comment 3: “MUC1–/– mice” and “MUC2–/– mice”: Please kindly check and revise the terms appropriately if necessary. Generally, MUC1 refers to a protein, and Muc1 refers to a gene.

Response: As suggested, we have changed the MUC gene designations to 'Muc1-/- mice' and 'Muc2-/- mice' (section 4.2).

Comment 4:  Abbreviations are required when they are reused. However, some abbreviations, such as NSAID, were introduced but not used elsewhere.

Response:  As suggested, abbreviations used only once, were removed from this paper (e.g. QOL, TPDs, and SMEDDS (introduction), NSAID (section5.1), PAMPA and SNEDDS (section5.2), ECM(section5.3)).